behaviour/physiology

sprint speed, locomotor performance, social hierarchy, feeding, zebrafish

**Author for correspondence:**
Frank Seebacher
e-mail: frank.seebacher@sydney.edu.au

# Social rank and not physiological capacity determines competitive success in zebrafish (*Danio rerio*)

Clare Miln, Ashley J. W. Ward and Frank Seebacher

School of Life and Environmental Sciences A08, University of Sydney, NSW 2006, Australia

AJWW, 0000-0003-0842-533X; FS, 0000-0002-2281-9311

Competition for resources shapes ecological and evolutionary relationships. Physiological capacities such as in locomotor performance can influence the fitness of individuals by increasing competitive success. Social hierarchy too can affect outcomes of competition by altering locomotor behaviour or because higher ranking individuals monopolize resources. Here, we tested the hypotheses that competitive success is determined by sprint performance or by social status. We show that sprint performance of individuals measured during escape responses (fast start) or in an accelerated sprint test did not correlate with realized sprint speed while competing for food within a social group of five fish; fast start and accelerated sprint speed were higher than realized speed. Social status within the group was the best predictor of competitive success, followed by realized speed. Social hierarchies in zebrafish are established within 7 days of their first encounter, and interestingly, there was a positive correlation between social status and realized speed 1 and 4 days after fish were placed in a group, but not after 7 days. These data indicate that physiological performance decreases in importance as social relationships are established. Also, maximal physiological capacities were not important for competitive success, but swimming speed changed with social context.

## 1. Introduction

Obtaining food is most fundamental for survival and fitness in heterotrophic organisms, and competition is a principal ecological and evolutionary driver [1]. The mechanisms that determine competitive success therefore also reveal the primary drivers

underlying ecological relationships [2]. Differences in locomotor capacity between individuals may explain success in scramble competition, although speed may trade off with food detection accuracy [3,4]. However, within a social group, individuals do not necessarily display their maximal physiological capacities, but alter behaviour depending on the social context [5,6]. Hence, locomotor capacity and social status may either independently or interactively determine competitive success in social animals. Our aim was to test this relationship between locomotor performance and social status in zebrafish (*Danio rerio*) competing for food.

Sprint performance is a repeatable trait that is thought to be under selection because of the advantages it confers for predator escape, prey capture, territorial defence and survival [7]. Sprints are characterized by short (less than 30 s) bursts of high-powered locomotion that are mediated by fast glycolytic muscle fibres [8]. In fish, maximal sprint performance is often measured as fast starts in response to an external stimulus eliciting an escape response [9]. Alternatively, sprint performance can be measured as the fastest speed attained during constant acceleration in a swimming flume [10]. However, animals under natural circumstances often do not move maximally, and performance is modulated by context [11]. For example, collared lizards used 25–90% of maximal sprint speed to forage, escape predators or defend territories [12].

Social relationships may influence locomotor behaviour [13], and they may alter the importance of locomotor performance for competitive success. It is possible that social status *per se* takes precedence over locomotor capacity in determining competitive success. In zebrafish, a social hierarchy among unfamiliar animals is established within 7 days [14]. This establishment period provides a suitable context to test for the relative importance of locomotor capacity and social status in determining competitive success, because social status would be relatively unimportant before a hierarchy is established. We therefore tested the hypothesis that (i) immediately after fish are placed in a group, social status is less important than sprint performance in determining competitive success within the group, but this relationship is reversed after 7 days. Because sprint performance may vary with context, we tested the hypothesis that (ii) sprint performance measured in individual fish in an arena or flume reflects sprint performance realized by individuals within the group. Lastly, sprint performance may determine social status or, vice versa, the realized speed in the group may be modified by social status as the hierarchy is established, so we tested the hypothesis that (iii) realized speed in the group is positively associated with social status.

# 2. Material and methods

## 2.1. Study animals

All animal handling and experimental procedures were conducted with the approval of the University of Sydney Animal Ethics Committee (approval number: 2016/982). Zebrafish, *Danio rerio*, form social hierarchies that are stable [15] and therefore are a good model to test hypotheses regarding the fitness consequences of social hierarchies. Adult short-fin zebrafish of mixed sex were purchased from a commercial supplier (Marine Fish Direct, Narellan, NSW, Australia) and housed in plastic tanks ($0.64 \times 0.45 \times 0.23$ m; 2–3 fish per litre) until the start of experiments (1–2 weeks). Each tank contained a sponge filter connected to an air pump (AC-9908; Resun, China), and we conducted a 30% water change once a week. Water temperature was 23°C ($\pm$ 0.5°C) during all experiments, the light cycle was 12 L : 12 D h, and fish were fed flake food (Wardley Total Tropical Flake Blend; Hartz Mountain Company, Secaucus NJ, USA) until satiation once daily. Fish were marked individually with a coloured elastomer tag injected subcutaneously (NorthWest Marine Technology Inc., Shaw Island, WA, USA). Each fish within an experimental group (see below) was marked with a different colour, which was readily distinguishable by direct observation and from video recordings. Fish were given at least 3 days to recover from the tagging procedure before starting experiments; after 3 days all fish fed and behaved normally (i.e. engaged in social behaviour and spent most of their time cruising near the water surface). Experimental fish were of mixed sex (with two or three fish of each sex per group) to provide a more natural context and to avoid bias from using a single sex only [16]. In zebrafish, both males and females engage in dominance behaviour [15]. All fish were weighed with an electronic balance (Navigator NV, Ohaus, USA), and photographed (with a Exilim EX-ZR200A, Casio, USA camera; electronic supplementary material, figure S1) to determine the standard length (in Tracker Video Analysis and Modelling Tool 4.01; Open Source Physics, www.opensourcephysics.org).

## 2.2. Sprint swimming performance

We measured swimming performance in all experimental fish (i.e.10 groups of five fish each, see below). There are different measurement techniques of sprint speed in the literature [17], and to avoid bias, we included two different measures. Hence, sprint speed was determined as an escape response and a constant acceleration test as described previously [10]. Escape responses were induced by a mechanical stimulus to the tail of the fish in a shallow tank ($405 \times 600$ mm, 30 mm water depth). We filmed the escape response from above at 120 frames s$^{-1}$ (with a GoPro Hero 6, San Mateo CA, USA), and a submerged 300 mm ruler served as a scale. We analysed videos (in Tracker Video Analysis and Modelling software, 4.01, Open Source Physics, www.opensourcephysics.org) using the fish centre of mass as the tracking point. Centre of mass was defined as the location at 0.35 body lengths from the tip to the snout [18]. The purpose of measuring fast starts is to determine maximal capacities. Hence, four escape responses were analysed for each fish and the fastest speed was used in the analysis, which we refer to as 'fast start' [17,19]. Escape responses were only used when the movement of fish was not impeded by the edge of the tank.

The second method consisted of determining the maximum attained speed ($U_{10}$) in a 10 s constant acceleration trial in a cylindrical, Blazka-style swimming flume (21 mm diameter $\times$ 150 mm length) [10]. Each flume was fitted tightly over the intake end of a submersible pump (12 V DC, iL500; Rule, Hertfordshire, UK) that drew water through the flume. The flow was adjusted by changing the voltage input to the pumps with a DC-regulated power supply (NP9615; Manson Engineering Industrial, Hong Kong). A flow meter (DigiFlow, 6710M, Savant Electronics, Taichung, Taiwan) fitted to the outlet of the pump measured flow velocity in real time. Before the test, fish swam at an initial flow rate of 0.05 m s$^{-1}$ for 30 s, after which water flow was continuously increased by an average of 0.05 m s$^{-1}$ every 1 s until fish could no longer maintain their position in the water column and fell back against a mesh net. The maximum flow speed was recorded when the fish was still holding its position in the water column and used as maximum sprint performance.

## 2.3. Competitive ability

To measure competitive success and social interactions, we assembled experimental groups ($n = 10$ groups) of five fish each. Fish within groups were size matched so that they were within 2 mm standard length of each other, and individuals had not been in contact with each other before the assembly of the groups. Each group was housed in a separate tank ($0.30 \times 0.21 \times 0.18$ m). For trials, all five fish per tank were placed underneath a perforated 1 l cylindrical container, which allowed visual inspection of the tank but prevented egress from the container. A piece of fish flake food (Wardley Total Tropical Flake Blend; Hartz Mountain Company, Secaucus, NJ) was then placed approximately 20 cm away from the perforated container. After 5 s, the container was removed allowing the fish to move freely around the tank. Fish were filmed (GoPro Hero 6, San Mateo CA, USA, at 120 frames s$^{-1}$) from just before the container was lifted, and the identity of the fish that obtained the food was recorded. This procedure was repeated 10 times consecutively per day for 7 days. In preliminary trials, we determined the best size of the food flake so that it was visible to the fish, but small enough for only one fish to obtain it. During filmed trials, water depth was lowered to 7 cm, and between trials fish groups remained in their tanks and water levels were increased to 15 cm when fish were not filmed. After the food trials each day fish were fed to satiation. From the videos, we also determined the speed at which fish travelled towards the food (realized speed). We defined realized speed as the maximum speed calculated across two video frames while a fish was moving after the container was lifted and until the food item was eaten by any fish. Note that the maximum speed attained during competitive trials was significantly correlated with the average speed (permutational analysis [see below] $p < 0.0001$), and this relationship did not differ between days ($p = 0.69$; electronic supplementary material, figure S2). We used maximum speed as a representation of realized speed because it provides a better comparison with sprint speed, which represents maximum physiological capacity.

## 2.4. Social interactions

Before each competitive trial, we filmed fish (with a GoPro Hero 6, San Mateo CA, USA, at 120 frames s$^{-1}$) moving freely and undisturbed in their home tank for 1 min; the first footage was taken immediately after fish were assembled into experimental groups on the first day. From videos, we recorded aggressive

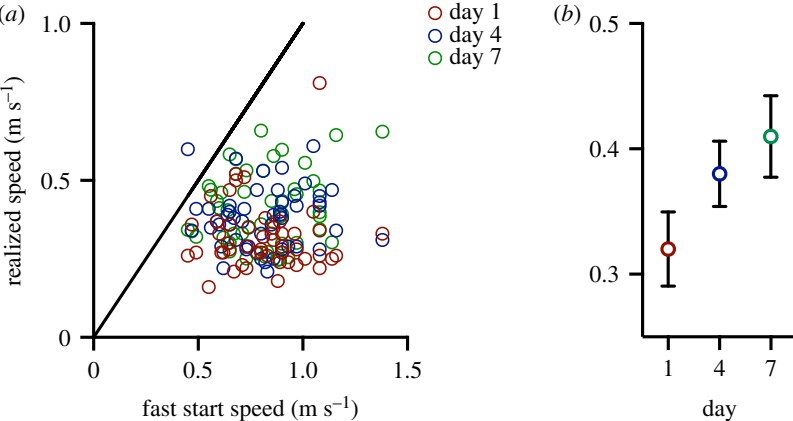

**Figure 1.** Sprint speed during competition and in an arena. Fast start speed measured individually in an arena was greater than the fastest speed realized by individuals when moving towards a food source within a group of five fish (realized speed) (*a*). There was no statistically significant relationship between fast start and realized speeds on day 1 (red circles), day 4 (blue circles) or day 7 (green circles). However, realized speed increased from day 1 to day 7 (*b*). Sample size was 10 groups of five fish on each day, the solid line shows $x = y$, and means $\pm$ 95% CI are shown in (*b*).

interactions between individuals. An aggressive interaction was defined as a charge by an individual towards another fish, or a nip towards another individual that elicited a response in the receiving individual. We used the method described in [20] to determine social scores, where a fish received a point (+1) for each incident where it was the aggressor, and a negative score (−1) for each aggressive interaction where it was the receiver. The total score for each fish per day was calculated by adding together the aggressor and receiver scores, and fish with higher scores were considered to be more dominant within the social hierarchy.

## 2.5. Statistical analyses

We analysed all data with permutational analyses in the R [21] package lmPerm [22]. Permutational analyses are a randomization procedure that use the data *per se* for statistical inference rather than assuming underlying distributions, which makes them more parsimonious and independent from underlying assumptions [23]. The *p*-value in permutational analyses is calculated as the number of randomized datasets that show the same or greater effect as the measured dataset, divided by the total number of permutations; the *p*-value therefore denotes the probability that the observed effects could be obtained randomly. Hence, permutational analyses do not have any associated test statistic and are often preferable to frequentist statistics [24]. The sample size was 50 fish in 10 groups of five fish each, and we give *p*-values and degrees of freedom ($p_{df}$).

In all models, we used 'tank' (=group) as a random blocking factor. In the event, 'tank' was not significant in any of the comparisons (all $p > 0.1$). To test the relationship between sprint speed and realized speed, we used realized speed as the dependent variable and fast start or $U_{10}$ speed and 'day' as the dependent variables. We used a fast start or $U_{10}$ speed, realized speed and social score as independent variables to test their influence on competitive success. Lastly, we tested whether realized speed influenced social score by using the social score as the dependent variable and realized speed and 'day' as independent variables. When 'day' was significant in the whole dataset, we repeated the analysis using subsets of data for each day to test on which days there were significant effects. We analysed data from days 1, 4 and 7 only. Effect sizes of different predictors on competitive success were calculated as Cohen's d, and 95% confidence intervals were determined by bootstrapping in the R package 'boot' [25].

## 3. Results

### 3.1. Fast starts or $U_{10}$ did not predict realized speed

Realized speed of individuals while moving towards the food within the group of five fish ($n = 10$ groups) was lower than fast start (figure 1*a*) and $U_{10}$ speed (electronic supplementary material, figure

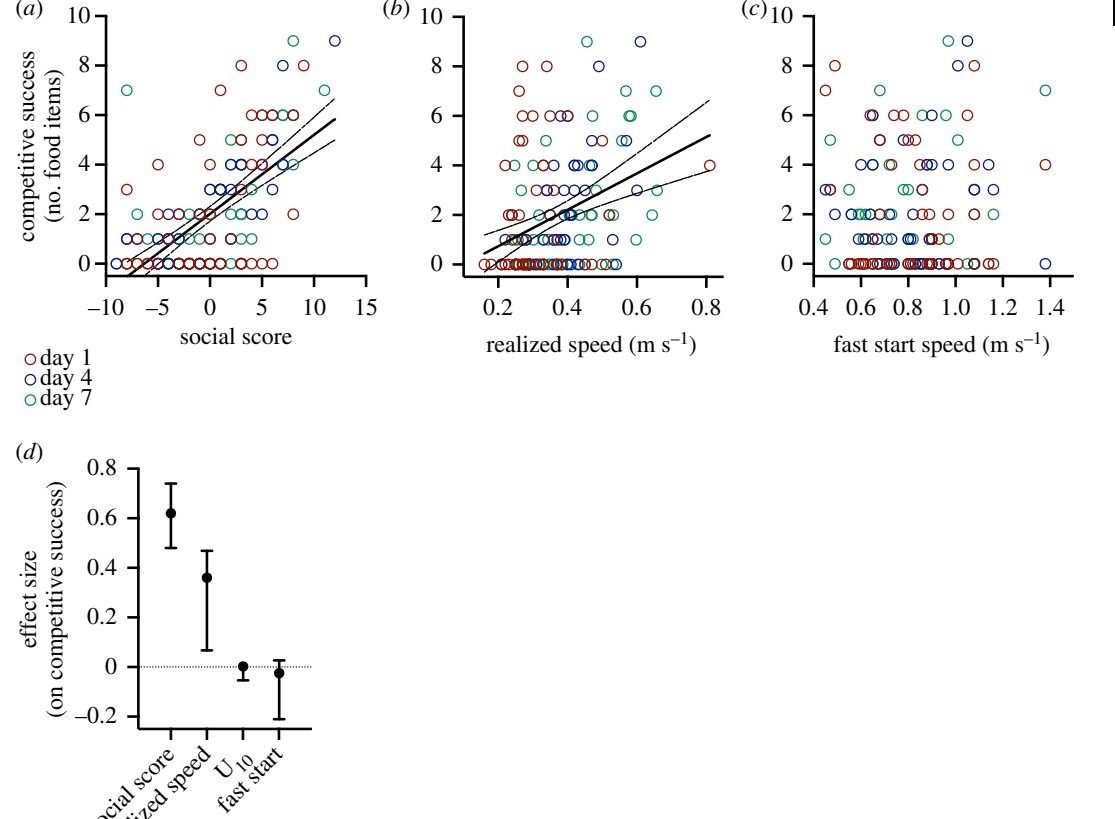

**Figure 2.** Predictors of competitive success. Social score (*a*) and speed realized by individuals when moving towards a food source within a group of five fish (realized speed) (*b*) were positively related to competitive success, measured as the number of food items obtained by an individual when competing in a group of five fish for 10 consecutive trials per day. Fast start speed was not associated with competitive success (*c*), and there were no differences between days (day 1 = red circles; day 4 = blue circles; day 7 = green circles). Social score had the greatest effect on competitive success (effect size) (*d*), followed by realized speed, but speed achieved during a 10 s accelerated spring test ($U_{10}$) or fast start speed did not have an effect on competitive success. Sample size was 10 groups of five fish on each day, and significant regression lines (± 95% CI) are shown ((*a*): $Y = -0.73 + 7.35x$, $R^2 = 0.13$; (*b*): $Y = 2.01 + 0.32x$, $R^2 = 0.39$).

S1a), and there was no statistically significant association between realized speed and fast start speed ($p_{1,137} = 0.3$) or $U_{10}$ speed ($p_{1,137} = 0.9$). However, between days 1 and 7 realized speed increased significantly by 0.19 m s$^{-1}$ on average ($p < 0.0001_{2,137}$; figure 1*b*). Additionally, fish with greater fast start speed reduced their realized speed to a disproportionally greater extent than fish with slower fast start speed ($p_{1,137} < 0.0001$), and this pattern was more pronounced on day 1 than on days 4 and 7 ($p_{2,137} < 0.0001$; electronic supplementary material, figure S3).

## 3.2. Realized speed and social score predicted competitive success

Realized speed ($p_{1,135} = 0.004$) and social score ($p_{1,135} < 0.0001$) had a positive effect on competitive success, but fast start speed ($p_{1,135} = 0.9$) and $U_{10}$ speed ($p_{1,135} = 0.1$) did not influence competitive success (figure 2; electronic supplementary material, figure S4). The effect size of social score on competitive success was greater than that of realized speed (95% CI; figure 2). There was no significant effect of day on these relationships ($p_{2,135} = 0.7$).

## 3.3. Social score was associated with realized speed only initially

There was a significant positive relationship between social score and realized speed ($p_{1,137} < 0.0001$), and there was a significant effect of day ($p_{2,137} < 0.0001$) (figure 3). The association between social score and realized speed was significant on day 1 ($p_{1,48} = 0.03$) and on day 4 ($p_{1,48} = 0.005$), but not on day 7 ($p_{1,48} = 0.3$).

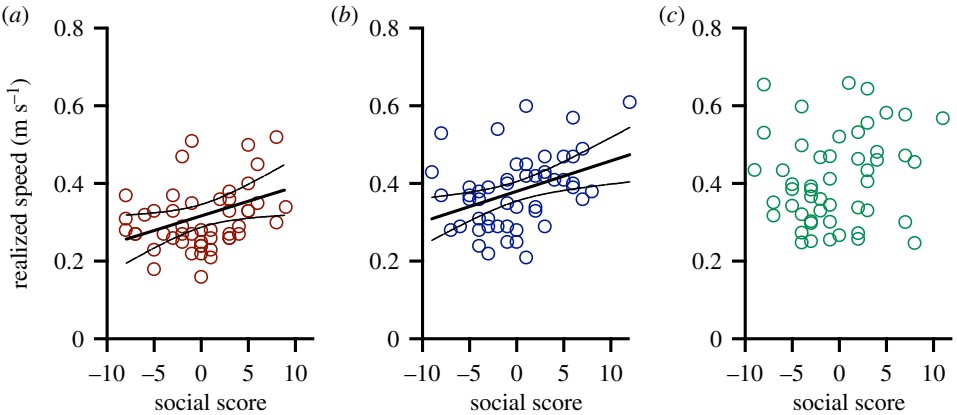

**Figure 3.** Social score predicted realized speed. There were weak but statistically significant positive relationships between realized speed and social score on day 1 (*a*) and day 4 (*b*), but not on day 7 (*c*). Sample size was 10 groups of five fish on each day, and significant regression lines ($\pm$ 95% CI) are shown ((*a*): $Y = -3.86 + 12.71x$, $R^2 = 0.096$; (*b*): $Y = -7.21 + 18.98x$, $R^2 = 0.15$).

## 4. Discussion

We have shown that position within a social hierarchy is the most important determinant of competitive success, although realized speed while competing does contribute to success. However, maximal physiological capacities did not lead to a competitive advantage, and realized speed while competing in a group was less than maximal performance.

Group living is advantageous because it protects individuals from predation, and information transfer between individuals in the group increases access to resources [26,27]. Group fission and fusion impacts population dynamics, and interactions between individuals and their relative positions within the group can influence fitness [28–30]. Social hierarchies within groups are common, and more dominant individuals within a group have greater competitive success and access to resources [31]. However, dominance can also come at the cost of increased stress and energy expenditure resulting from increased aggression to maintain social position [32]. Physiological capacities are often associated with group dynamics [4,33], and increased capacities may promote success in aggressive encounters and dominance [34].

The consequences of social hierarchies are particularly important when competing for limited resources such as food, which can lead to differences in fitness. However, even though subordinate individuals experience restricted access to resources, they remain within the group because of the benefits of group living [35]. It is therefore advantageous for individuals and the group as a whole to reduce levels of aggression within a group. Individuals may therefore adjust their behaviour to minimize the costs and maximize the benefits of group living [36]. Subordinates within a group may suppress their potential physical performance to avoid challenging dominant individuals and thereby avoid aggressive encounters [37]. We show that even while being established, social hierarchy plays an important role in food acquisition. Despite having the potential swimming capacity to win limited items of food, we showed that social status within the group had a greater effect on success to obtain food. Subordinates within a group may benefit from forgoing opportunities to obtain food to reduce conflict with the more dominant fish [38].

It is interesting that animals rarely perform maximally under natural or naturalistic situations [11,12], which is likely to be the result of trade-offs with other fitness-related traits [39]. Fish with greater fast start speeds showed a disproportionally greater reduction to realized speed than fish with lower fast start speeds, indicating that fish tended to conform to the group regardless of their physiological capacities. The observation that realized speed increased during the week indicates that the stimulus to compete did not decrease during that period. Movement while competing may be interpreted within a conceptual framework that is similar to economic escape theory. Economic escape theory posits that predator escape responses are determined by the trade-off between the costs of fleeing (e.g. missed opportunities and energetic investment) and the costs of not fleeing (loss of lifetime fitness) [40]. While competing for food, the cost of obtaining food would include increased swimming speed and energetic investment, and the increased likelihood of subsequent conflict with others in the group. The cost of not obtaining food would be a decrement in nutrition. These two costs would need to be

traded off while competing in a group. Dominant individuals may minimize swim speed to save energy while still obtaining food, and subordinate individuals could adjust swim speed according to that of more dominant fish. Also, the stimulus to escape a predator (fast starts) may be far stronger than to obtain food. These dynamics could explain why realized speeds are lower than maximal speeds, and why the social score is related to the realized speed at least early when hierarchies are established.

Ethics. All animal handling and experimental procedures were conducted with the approval of the University of Sydney Animal Ethics Committee (approval no.: 2016/982).

Data accessibility. Data has been deposited in Dryad (https://doi.org/10.5061/dryad.v15dv41tr) [41].

Authors' contributions. F.S., C.M. and A.J.W.W. designed the study; C.M. conducted experiments; F.S. wrote the manuscript and analysed data; F.S., C.M. and A.J.W.W. edited the manuscript

Competing interests. We declare we have no competing interests

Funding. This work was supported by an Australian Research Council Discovery grant (DP180103036) to F.S.

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
