## [Peer Review File · Royal Society Open Science]

Review History

RSOS-201640.R0 (Original submission)

Review form: Reviewer 1

Is the manuscript scientifically sound in its present form?

Yes

Are the interpretations and conclusions justified by the results?

Yes

Is the language acceptable?

Yes

Do you have any ethical concerns with this paper?

No

Have you any concerns about statistical analyses in this paper?

Yes

Recommendation?

Major revision is needed (please make suggestions in comments)

Comments to the Author(s)

See the attached file for my comments to the authors (Appendix A).

Review form: Reviewer 2

Is the manuscript scientifically sound in its present form?

Yes

Are the interpretations and conclusions justified by the results?

Yes

Is the language acceptable?

Yes

Do you have any ethical concerns with this paper?

No

Have you any concerns about statistical analyses in this paper?

No

Recommendation?

Major revision is needed (please make suggestions in comments)

Comments to the Author(s)

Summary

In this manuscript, Miln et al. present an experimental study in which they aimed to identify the determinants of competitive success in zebrafish. By measuring predator escape responses and competitive foraging speeds, the authors show that competitive success (defined as being the first fish to reach a piece of food) is determined by an individual's ranking in the social hierarchy rather than their ability to swim fast. These results suggest that, in gregarious species, individuals may modulate their foraging speeds based on their social status, thus trading-off the benefits and opportunity costs of group living.

Overall, this manuscript was very well written and enjoyable to read (nice job!). I have relatively few concerns about the paper, although I do see some opportunities for improvement. I have split my feedback into major and minor comments below.

Major Comments:

- Lines 135-137: "Four escape responses were analysed for each fish and the fastest speed was used in the analysis, which we refer to as 'fast start'."

Can the authors please clarify why they elected to only use the fastest escape response for each individual, and exclude the other three measurements? It seems to me that a similar analysis could be conducted that makes use of all speed measurements for each fish (for example, by including individual as a random effect), or (less preferably) by taking the average speed across the four trials. Presumably, the escape response is not perfectly repeatable, and so there is some (possibly substantial) degree of within-individual variation. Does the analysis hold if, for example, the slowest times are used instead?

- Results

Can the authors please report all the relevant statistical information to support their results? In my view, reporting only p-values in the absence of test statistics, sample sizes, and degrees of freedom, is insufficient.

Minor Comments:

- Lines 39-40: "Here we tested the alternative hypotheses that competitive success is determined by sprint performance or by social status."

Are these two hypotheses necessarily mutually-exclusive alternatives? What if, for example, the fastest swimmers have the highest social status, and so the two are actually confounded with each other (i.e., hypothesis 3 as outlined in the introduction)?

- Lines 44-45: "Social hierarchies in zebrafish are established within seven days..."

Can the authors clarify what this means? Within seven days of what (presumably group formation)?

- Lines 89-90: "... and it may alter the importance of locomotor..."

Not clear what "it" is referring to here (social relationships or locomotor behaviour). Can the authors please clarify? If "it" refers to social relationships, I recommend replacing "it" with "they".

- Lines 95-96: "We therefore tested the hypotheses that i)..."

Replace the plural "hypotheses" with the singular "hypothesis".

- Lines 123;124: "... and photographed to determine standard length..."

I know that this is a standard procedure, but I think it might be useful to include an example photo (or examples) in the supplementary materials to show how length was measured.

- Line 129: "... previously[10]."

Need a space before the reference.

- Lines 159-160: "A piece of fish flake food was then placed approximately 20 cm away from the perforated container. After five seconds the contained was removed..."

Is it possible the a fish's position within the holding cylinder and thus (Euclidean) distance to the piece of food prior to the container being removed affected which fish got the food? That is, do fish who start the trial closer to the food have a higher likelihood of getting the food (i.e., not because they are better competitors per se)? Of course, repeated sampling (10 trials per group per day) should randomize fish's starting positions, so perhaps this is not a concern.

- Lines 170-172: ""

Similar to my comment above about escape responses, I'm slightly unclear why the maximum speed was used as a proxy for realized speed here. Can the authors please clarify? It seems to me that average speed would be a much more suitable estimate of realized speed (i.e., if speed is normally distributed, the time it takes to travel between two points is much more dependent on the average speed than short-duration bursts of maximum speed that occur at the extremes). Although, perhaps given the relatively short distances being measured the average and maximum are similar to each other?

- Lines 207-208: "However, realised speed increased significantly ($p < 0.0001$) from day 1 to day 7."

I recommend re-structuring this sentence (and others in the results) to refocus the results on the biology and less on arbitrary conclusions about significance/non-significance. For example, what was the effect size or average increase in speed between days 1 and 7? For example: "Between days 1 and 7, realized speeds increased by an average of XX m/s (test-stat, df, $p < 0.0001$)."

- Lines 251-252: "It is interesting that animals rarely perform maximally under natural or naturalistic situations, which is likely to be the result of trade-offs with other fitness-related traits."

Is it possible that fish with different maximum capacities adjusted their realized speeds in different ways? From a competitive perspective, it's not essential that a fish travels at its maximum speed, but that instead that it has the fastest speed among the group of competitors. Is there any indication that, for example, those fish with fast start speeds modulate their realized speed in response to the realized speeds of others in the group? In this way, fish who *can* travel fast will appear to have reduced their realized speeds so they are only slightly faster than the next fastest fish. In contrast, fish who have slower start speeds might express relatively similar realized speeds because they need to travel near their maximum in order to compete (i.e., a conditional correlation).

- Lines 254-255: "Movement while competing may be interpreted within a conceptual framework that is similar to Economic Escape Theory."

Relatedly, perhaps similar theory also explains the lack of a significant correlation between maximum escape speed (in response to predator stimulus) and realized speed (in response to food). The outcomes of these two contexts/scenarios are fundamentally unbalanced (i.e., the very serious risk of death vs. the not-so-serious risk of missing out on a single foraging opportunity; "[swimming] for your life vs. [swimming] for your dinner").

Review form: Reviewer 3

Is the manuscript scientifically sound in its present form?

No

Are the interpretations and conclusions justified by the results?

No

Is the language acceptable?

Yes

Do you have any ethical concerns with this paper?

No

Have you any concerns about statistical analyses in this paper?

Yes

Recommendation?

Major revision is needed (please make suggestions in comments)

Comments to the Author(s)

The study is an investigation of the relationship between traits related to physiological capacity as swimming performance and competitive success in zebrafish. It deals with the premise that social ranking can impact competitive success too and the study indicates that physiological performance decreases in importance as social relationships are established.

While this is a relevant and interesting question to address, the study itself seems more preliminary at this stage. The conclusions drawn from the results are not satisfactory, as these are not currently sufficiently supported by the analysis and results presented. Details provided in the methods section are rather insufficient and the statistical analyses also need to be improved. It is

well known that biological factors such as the sex of individuals within groups can be important in the development of social ranking. In zebrafish too, studies have shown the role of sex in the development of dominance (e.g. Paull et al. 2010; Dahlbom et al. 2012). This has not been addressed or discussed in the study. The results should be discussed in the context of what is known across other similar group-living species.

The manuscript needs major revision before it can be recommended for acceptance. Specific comments are provided below:

Material and Methods: Line 111. How many fish were kept in each tank? How long were they kept there before starting experiments?

Line 121. How do you know 3 days are enough for fish to recover after elastomer tagging? How was it ensured that elastomer tagging did not affect stress levels in individuals?

Perhaps it would help to have control tests where individuals with and without tags were compared for their sprint performance or the same individuals can be tested for sprint performance before and after tagging?

Line 150. How many individuals were tested for their sprint performance? Were they all males, all females, or both?

Line 153. As mentioned above, what was the sex of the individuals' groups? Where these mixed-sex groups?

Line 181. "system of? " Not clear. Please rewrite.

Line 185. it is well believed that there are differences in aggressive interactions when there are same-sex or mixed-sex groups.

So it is important to address the role of the sex of individuals to understand the formation of social hierarchy.

Statistical Analysis: Line 194. Was a random factor included in the model? As the same fish are tested across days, fish id would be used as a random factor in the model.

Results: Lines 204-208. It is difficult to understand the results of the study in the current version of the manuscript. Please provide sample sizes or df for your statistical analyses. Estimate values for the model should also be given.

Line 210. "Realised speed and social score predicted competitive success"

Results need to be reported with the estimate, df, etc. of the model

Line 216. What are the model R square, df, etc. for these analyses? Based on the Figure provided, the relationship is a weak positive one.

Discussion: Line 223. This statement is not convincing, as the results presented do not necessarily prove social hierarchy as being the most important determinant of competitive success. The figures also do not show a strong enough relationship. This statement should be rewritten.

Line 228-230. Ending a sentence with "for example" sounds strange. This statement needs to be reworded.

Line 262-266. This needs to be checked, because, details of the results are not provided and the figure at least only seems to show a weakly positive relationship.

Decision letter (RSOS-201640.R0)

Dear Dr Seebacher

The Editors assigned to your paper RSOS-201640 "Social rank and not physiological capacity determines competitive success in zebrafish (*Danio rerio*)" have made a decision based on their reading of the paper and any comments received from reviewers.

Regrettably, in view of the reports received, the manuscript has been rejected in its current form. However, a new manuscript may be submitted which takes into consideration these comments.

We invite you to respond to the comments supplied below and prepare a resubmission of your manuscript. Below the referees' and Editors' comments (where applicable) we provide additional requirements. We provide guidance below to help you prepare your revision.

Please note that resubmitting your manuscript does not guarantee eventual acceptance, and we do not generally allow multiple rounds of revision and resubmission, so we urge you to make every effort to fully address all of the comments at this stage. If deemed necessary by the Editors, your manuscript will be sent back to one or more of the original reviewers for assessment. If the original reviewers are not available, we may invite new reviewers.

Please resubmit your revised manuscript and required files (see below) no later than 16-May-2021. Note: the ScholarOne system will 'lock' if resubmission is attempted on or after this deadline. If you do not think you will be able to meet this deadline, please contact the editorial office immediately.

Please note article processing charges apply to papers accepted for publication in Royal Society Open Science (<https://royalsocietypublishing.org/rsos/charges>). Charges will also apply to papers transferred to the journal from other Royal Society Publishing journals, as well as papers submitted as part of our collaboration with the Royal Society of Chemistry (<https://royalsocietypublishing.org/rsos/chemistry>). Fee waivers are available but must be requested when you submit your manuscript (<https://royalsocietypublishing.org/rsos/waivers>).

Thank you for submitting your manuscript to Royal Society Open Science and we look forward to receiving your resubmission. If you have any questions at all, please do not hesitate to get in touch.

on behalf of Dr Punidan Jeyasingh (Associate Editor) and Kevin Padian (Subject Editor)
openscience@royalsociety.org

Editor comments:

Thanks for your submission, which has generally interested our reviewers and AE. All have reservations and of different kinds, and so mainly for the opportunity to give you more than our standard three weeks to revise under a "major revision" decision, I'm electing a "reject/resub" so that you won't feel rushed. Because you did not get a round of external review with Proc B, we will ask our reviewers if they will be willing to look over a resubmission.

One reviewer commented at some point on our long form, but did not repeat it in the comments to the authors, the following question and I bring it to your attention because it does not seem like a privileged comment. The use of an Excel file can be problematic as a suitable long-term storage form and so the reviewer wonders if you can also submit a .csv format with the values

separated by commas, one file each for the data and metadata, and also the scripts for your code. Thanks very much.

Associate Editor Comments to Author (Dr Punidan Jeyasingh):

This study delves into poorly understood interactions between individual ability and social status in determining competition for food resources. The manuscript was assessed by three experts. All three were enthusiastic about the study motivation and approach. Nevertheless, all three were uniformly critical of the analyses employed (in addition to a few other key points unique to each expert). I felt the reviews were fair, clear, and constructive. With much gratitude to the expert reviewers, I invite the authors to address/incorporate these comments. I look forward to reading a revised version.

Reviewer comments to Author:

Reviewer: 1

Comments to the Author(s)

See the attached file for my comments to the authors.

Reviewer: 2

Comments to the Author(s)

Summary

In this manuscript, Miln et al. present an experimental study in which they aimed to identify the determinants of competitive success in zebrafish. By measuring predator escape responses and competitive foraging speeds, the authors show that competitive success (defined as being the first fish to reach a piece of food) is determined by an individual's ranking in the social hierarchy rather than their ability to swim fast. These results suggest that, in gregarious species, individuals may modulate their foraging speeds based on their social status, thus trading-off the benefits and opportunity costs of group living.

Overall, this manuscript was very well written and enjoyable to read (nice job!). I have relatively few concerns about the paper, although I do see some opportunities for improvement. I have split my feedback into major and minor comments below.

Major Comments:

- Lines 135-137: "Four escape responses were analysed for each fish and the fastest speed was used in the analysis, which we refer to as 'fast start'."

Can the authors please clarify why they elected to only use the fastest escape response for each individual, and exclude the other three measurements? It seems to me that a similar analysis could be conducted that makes use of all speed measurements for each fish (for example, by including individual as a random effect), or (less preferably) by taking the average speed across the four trials. Presumably, the escape response is not perfectly repeatable, and so there is some (possibly substantial) degree of within-individual variation. Does the analysis hold if, for example, the slowest times are used instead?

- Results

Can the authors please report all the relevant statistical information to support their results? In my view, reporting only p-values in the absence of test statistics, sample sizes, and degrees of freedom, is insufficient.

Minor Comments:

- Lines 39-40: "Here we tested the alternative hypotheses that competitive success is determined by sprint performance or by social status."

Are these two hypotheses necessarily mutually-exclusive alternatives? What if, for example, the fastest swimmers have the highest social status, and so the two are actually confounded with each other (i.e., hypothesis 3 as outlined in the introduction)?

- Lines 44-45: "Social hierarchies in zebrafish are established within seven days..."

Can the authors clarify what this means? Within seven days of what (presumably group formation)?

- Lines 89-90: "..., and it may alter the importance of locomotor..."

Not clear what "it" is referring to here (social relationships or locomotor behaviour). Can the authors please clarify? If "it" refers to social relationships, I recommend replacing "it" with "they".

- Lines 95-96: "We therefore tested the hypotheses that i)..."

Replace the plural "hypotheses" with the singular "hypothesis".

- Lines 123;124: "... and photographed to determine standard length..."

I know that this is a standard procedure, but I think it might be useful to include an example photo (or examples) in the supplementary materials to show how length was measured.

- Line 129: "... previously[10]."

Need a space before the reference.

- Lines 159-160: "A piece of fish flake food was then placed approximately 20 cm away from the perforated container. After five seconds the contained was removed..."

Is it possible the a fish's position within the holding cylinder and thus (Euclidean) distance to the piece of food prior to the container being removed affected which fish got the food? That is, do fish who start the trial closer to the food have a higher likelihood of getting the food (i.e., not because they are better competitors per se)? Of course, repeated sampling (10 trials per group per day) should randomize fish's starting positions, so perhaps this is not a concern.

- Lines 170-172: ""

Similar to my comment above about escape responses, I'm slightly unclear why the maximum speed was used as a proxy for realized speed here. Can the authors please clarify? It seems to me that average speed would be a much more suitable estimate of realized speed (i.e., if speed is normally distributed, the time it takes to travel between two points is much more dependent on the average speed than short-duration bursts of maximum speed that occur at the extremes). Although, perhaps given the relatively short distances being measured the average and maximum are similar to each other?

- Lines 207-208: "However, realised speed increased significantly ($p < 0.0001$) from day 1 to day 7."

I recommend re-structuring this sentence (and others in the results) to refocus the results on the biology and less on arbitrary conclusions about significance/non-significance. For example, what was the effect size or average increase in speed between days 1 and 7? For example: "Between days 1 and 7, realized speeds increased by an average of XX m/s (test-stat, df, $p < 0.0001$)."

- Lines 251-252: "It is interesting that animals rarely perform maximally under natural or naturalistic situations, which is likely to be the result of trade-offs with other fitness-related traits."

Is it possible that fish with different maximum capacities adjusted their realized speeds in different ways? From a competitive perspective, it's not essential that a fish travels at it's maximum speed, but that instead that it has the fastest speed among the group of competitors. Is there any indication that, for example, those fish with fast start speeds modulate their realized

speed in response to the realized speeds of others in the group? In this way, fish who *can* travel fast will appear to have reduced their realized speeds so they are only slightly faster than the next fastest fish. In contrast, fish who have slower start speeds might express relatively similar realized speeds because they need to travel near their maximum in order to compete (i.e., a conditional correlation).

- Lines 254-255: "Movement while competing may be interpreted within a conceptual framework that is similar to Economic Escape Theory."

Relatedly, perhaps similar theory also explains the lack of a significant correlation between maximum escape speed (in response to predator stimulus) and realized speed (in response to food). The outcomes of these two contexts/scenarios are fundamentally unbalanced (i.e., the very serious risk of death vs. the not-so-serious risk of missing out on a single foraging opportunity; "[swimming] for your life vs. [swimming] for your dinner").

Reviewer: 3

Comments to the Author(s)

The study is an investigation of the relationship between traits related to physiological capacity as swimming performance and competitive success in zebrafish. It deals with the premise that social ranking can impact competitive success too and the study indicates that physiological performance decreases in importance as social relationships are established.

While this is a relevant and interesting question to address, the study itself seems more preliminary at this stage. The conclusions drawn from the results are not satisfactory, as these are not currently sufficiently supported by the analysis and results presented. Details provided in the methods section are rather insufficient and the statistical analyses also need to be improved. It is well known that biological factors such as the sex of individuals within groups can be important in the development of social ranking. In zebrafish too, studies have shown the role of sex in the development of dominance (e.g. Paull et al. 2010; Dahlbom et al. 2012). This has not been addressed or discussed in the study. The results should be discussed in the context of what is known across other similar group-living species.

The manuscript needs major revision before it can be recommended for acceptance. Specific comments are provided below:

Material and Methods: Line 111. How many fish were kept in each tank? How long were they kept there before starting experiments?

Line 121. How do you know 3 days are enough for fish to recover after elastomer tagging? How was it ensured that elastomer tagging did not affect stress levels in individuals?

Perhaps it would help to have control tests where individuals with and without tags were compared for their sprint performance or the same individuals can be tested for sprint performance before and after tagging?

Line 150. How many individuals were tested for their sprint performance? Were they all males, all females, or both?

Line 153. As mentioned above, what was the sex of the individuals' groups? Where these mixed-sex groups?

Line 181. "system of?" Not clear. Please rewrite.

Line 185. it is well believed that there are differences in aggressive interactions when there are same-sex or mixed-sex groups.

So it is important to address the role of the sex of individuals to understand the formation of social hierarchy.

Statistical Analysis: Line 194. Was a random factor included in the model? As the same fish are tested across days, fish id would be used as a random factor in the model.

Results: Lines 204-208. It is difficult to understand the results of the study in the current version of the manuscript. Please provide sample sizes or df for your statistical analyses. Estimate values for the model should also be given.

Line 210. "Realised speed and social score predicted competitive success"

Results need to be reported with the estimate, df, etc. of the model

Line 216. What are the model R square, df, etc. for these analyses? Based on the Figure provided, the relationship is a weak positive one.

Discussion: Line 223. This statement is not convincing, as the results presented do not necessarily prove social hierarchy as being the most important determinant of competitive success. The figures also do not show a strong enough relationship. This statement should be rewritten.

Line 228-230. Ending a sentence with "for example" sounds strange. This statement needs to be reworded.

Line 262-266. This needs to be checked, because, details of the results are not provided and the figure at least only seems to show a weakly positive relationship.

===PREPARING YOUR MANUSCRIPT===

===PREPARING YOUR REVISION IN SCHOLARONE===

Author's Response to Decision Letter for (RSOS-201640.R0)

See Appendix B.

RSOS-210146.R0

Review form: Reviewer 1

Is the manuscript scientifically sound in its present form?

Yes

Are the interpretations and conclusions justified by the results?

Yes

Is the language acceptable?

Yes

Do you have any ethical concerns with this paper?

No

Have you any concerns about statistical analyses in this paper?

No

Recommendation?

Accept as is

Comments to the Author(s)

This manuscript is well-written, and all of my previous comments were either enacted satisfactorily in the revised manuscript or the authors provided a satisfactory defense for not incorporating my comments. The manuscript provides a novel, scientifically sound contribution to the field of biology that was well-written and a pleasure to read.

Review form: Reviewer 2

Is the manuscript scientifically sound in its present form?

Yes

Are the interpretations and conclusions justified by the results?

Yes

Is the language acceptable?

Yes

Do you have any ethical concerns with this paper?

No

Have you any concerns about statistical analyses in this paper?

No

Recommendation?

Accept as is

Comments to the Author(s)

Summary

Miln et al. present an experimental study in which they aimed to identify the determinants of competitive success in zebrafish. By measuring predator escape responses and competitive foraging speeds, the authors show that competitive success is determined by an individual's ranking in the social hierarchy rather than their ability to swim fast.

By my reading, the authors have done an excellent job of addressing my prior comments (as well as those from the other reviewers). In particular, I think the addition of analyses comparing (1) average vs. maximum realized speeds, and (2) the effect of social position on the relationship between start speed and realized speed have both strengthened the overall methodology and provide further support for the results.

I've enjoyed reviewing the manuscript, and wish the authors luck with its publication! I provide only a few minor comments below for their consideration.

(Line numbers correspond to the version with track-changes incorporated/removed.)

Minor Comments

- Lines 139, 170, 188: "We filmed... at 120 frames per second (with a GoPro Hero 6, San Mateo CA, USA)..."

This information is repeated several times in different forms throughout the methods section. For succinctness, I recommend revising to clarify that all video recordings were made at the same frame rate, and removing all but the first mention of the manufacturer and location.

- Lines 227-228: "...U10 speed (Supplementary Fig. 1a)."

I believe this should actually refer to supplementary Fig. S3a, rather than Fig. S1a (which doesn't exist). That said, the corresponding figure is currently presented as S4 in the supplementary materials. Supplementary items should be ordered in the same order they are referenced in the manuscript (so current S3 and S4 should be swapped).

- Line 233: "... Supplementary Fig. S3)."

See comment above. The supplementary materials should be re-ordered so this is Fig. S4 rather than Fig. S3.

- Line 238: "... (Fig. 2; Supplementary Fig. S4)."

See comment above.

Review form: Reviewer 3

Is the manuscript scientifically sound in its present form?

Yes

Are the interpretations and conclusions justified by the results?

Yes

Is the language acceptable?

Yes

Do you have any ethical concerns with this paper?

No

Have you any concerns about statistical analyses in this paper?

No

Recommendation?

Accept as is

Comments to the Author(s)

The authors have now addressed each of my concerns satisfactorily. The manuscript is now written with better clarity and the discussion also incorporates explanations of the results obtained.

Further explanations on the statistical analysis and justifications have now been provided in the manuscript. I recommend this manuscript for acceptance.

Decision letter (RSOS-210146.R0)

Dear Dr Seebacher,

I am pleased to inform you that your manuscript entitled "Social rank and not physiological capacity determines competitive success in zebrafish (*Danio rerio*)" is now accepted for publication in Royal Society Open Science.

Kind regards,

Royal Society Open Science Editorial Office
Royal Society Open Science

on behalf of Dr Punidan Jeyasingh (Associate Editor) and Kevin Padian (Subject Editor)
openscience@royalsociety.org

Associate Editor Comments to Author (Dr Punidan Jeyasingh):

Associate Editor

Comments to the Author:

I thank the authors for a thorough revision. The revision was reassessed by 3 experts, all of whom were satisfied with the way their comments were addressed (note that there are a few remaining minor points). This manuscript is much improved, and I am happy to recommend it for publication. I thank the authors and reviewers for making the review process efficient and constructive.

Reviewer comments to Author:

Reviewer: 1

Comments to the Author(s)

This manuscript is well-written, and all of my previous comments were either enacted satisfactorily in the revised manuscript or the authors provided a satisfactory defense for not incorporating my comments. The manuscript provides a novel, scientifically sound contribution to the field of biology that was well-written and a pleasure to read.

Reviewer: 2

Comments to the Author(s)

Summary

Miln et al. present an experimental study in which they aimed to identify the determinants of competitive success in zebrafish. By measuring predator escape responses and competitive foraging speeds, the authors show that competitive success is determined by an individual's ranking in the social hierarchy rather than their ability to swim fast.

By my reading, the authors have done an excellent job of addressing my prior comments (as well as those from the other reviewers). In particular, I think the addition of analyses comparing (1) average vs. maximum realized speeds, and (2) the effect of social position on the relationship between start speed and realized speed have both strengthened the overall methodology and provide further support for the results.

I've enjoyed reviewing the manuscript, and wish the authors luck with its publication! I provide only a few minor comments below for their consideration.

(Line numbers correspond to the version with track-changes incorporated/removed.)

Minor Comments

- Lines 139, 170, 188: "We filmed... at 120 frames per second (with a GoPro Hero 6, San Mateo CA, USA)..."

This information is repeated several times in different forms throughout the methods section. For succinctness, I recommend revising to clarify that all video recordings were made at the same frame rate, and removing all but the first mention of the manufacturer and location.

- Lines 227-228: "...U10 speed (Supplementary Fig. 1a)."

I believe this should actually refer to supplementary Fig. S3a, rather than Fig. S1a (which doesn't exist). That said, the corresponding figure is currently presented as S4 in the supplementary materials. Supplementary items should be ordered in the same order they are referenced in the manuscript (so current S3 and S4 should be swapped).

- Line 233: "... Supplementary Fig. S3)."

See comment above. The supplementary materials should be re-ordered so this is Fig. S4 rather than Fig. S3.

- Line 238: "... (Fig. 2; Supplementary Fig. S4)."

See comment above.

Reviewer: 3

Comments to the Author(s)

The authors have now addressed each of my concerns satisfactorily. The manuscript is now written with better clarity and the discussion also incorporates explanations of the results obtained.

Further explanations on the statistical analysis and justifications have now been provided in the manuscript. I recommend this manuscript for acceptance.

Appendix A

Miln et al, 2020 review

Positive comments

- I thought the experiment addressed novel ideas in the field by analyzing how sociality can impact organismal performance and thus impact competition.
- The paper was succinct, but with sufficient detail.
- The hypotheses were clearly laid out in the introduction, and it was easy to follow the experimental methods to test those hypotheses, and the results of those tests.

Negative comments

Minor comments

- Line 128: "...an escape response and an constant acceleration test..." should be changed to "an escape response and *a* constant acceleration test..."
- Under the statistical analyses section, you should cite R and the correct version number used in your analyses.
- Line 249: I think "obtained" should be changed to "obtain".
- In line 188, is there a reason why you implemented permutational linear models, as opposed to ordinary linear models? Were you expecting to have large deviations from the normal distribution in your responses? If so, I would include that information to justify its use.

Major comments

- I am concerned with your statistical approach for analyzing the influence of realized speed on social score. By only analyzing the data on days 1, 4, and 7 in this analysis, you could introduce confirmation bias into your results, and artificially inflate or deflate effects in your models. I think you should include all of your data in all analyses.
- In the sprint swimming performance analysis section, I believe it would be more statistically sound to use the average of the four escape responses, rather than the fastest one.
- Was there a relationship between the fast start speed and the U₁₀ speed? If there is a correlation between those two variables, you could either just use one, or create a composite variable. From the paper, it is unclear to me why both of these phenotypes were measured for "sprint speed".

Appendix B

Responses to editor and reviewer comments

Editor comments:

Thanks for your submission, which has generally interested our reviewers and AE. All have reservations and of different kinds, and so mainly for the opportunity to give you more than our standard three weeks to revise under a "major revision" decision, I'm electing a "reject/resub" so that you won't feel rushed. Because you did not get a round of external review with Proc B, we will ask our reviewers if they will be willing to look over a resubmission.

One reviewer commented at some point on our long form, but did not repeat it in the comments to the authors, the following question and I bring it to your attention because it does not seem like a privileged comment. The use of an Excel file can be problematic as a suitable long-term storage form and so the reviewer wonders if you can also submit a .csv format with the values separated by commas, one file each for the data and metadata, and also the scripts for your code. Thanks very much.

RESPONSE: we have now added .csv files of the descriptive metadata and the complete data set, as well as the ImPerm code to Dryad. Thanks also for the extra time!

Associate Editor Comments to Author (Dr Punidan Jeyasingh):

This study delves into poorly understood interactions between individual ability and social status in determining competition for food resources. The manuscript was assessed by three experts. All three were enthusiastic about the study motivation and approach. Nevertheless, all three were uniformly critical of the analyses employed (in addition to a few other key points unique to each expert). I felt the reviews were fair, clear, and constructive. With much gratitude to the expert reviewers, I invite the authors to address/incorporate these comments. I look forward to reading a revised version.

Reviewer comments to Author:

Reviewer: 1

Comments to the Author(s)

See the attached file for my comments to the authors.

Miln et al, 2020 review

Positive comments

- I thought the experiment addressed novel ideas in the field by analyzing how sociality can impact organismal performance and thus impact competition.
- The paper was succinct, but with sufficient detail.
- The hypotheses were clearly laid out in the introduction, and it was easy to follow the experimental methods to test those hypotheses, and the results of those tests.

Negative comments

Minor comments

- Line 128: "...an escape response and an constant acceleration test..." should be changed to "an escape response and a constant acceleration test..."

RESPONSE: corrected

- Under the statistical analyses section, you should cite R and the correct version number used in your analyses.

RESPONSE: We now cite the appropriate R version

- Line 249: I think "obtained" should be changed to "obtain".

RESPONSE: corrected

- In line 188, is there a reason why you implemented permutational linear models, as opposed to ordinary linear models? Were you expecting to have large deviations from the normal distribution in your responses? If so, I would include that information to justify its use.

RESPONSE: we have now provided more information and references on permutational analyses explaining why we think that they are preferable to frequentist statistics

Major comments

- I am concerned with your statistical approach for analyzing the influence of realized speed on social score. By only analyzing the data on days 1, 4, and 7 in this analysis, you could introduce confirmation bias into your results, and artificially inflate or deflate effects in your models. I think you should include all of your data in all analyses.

RESPONSE: Strictly speaking, to test our hypotheses we required only day 1 and day 7 (i.e. before and after hierarchies were established), but we added day 4 for extra resolution. Quite honestly, for that reason we never extracted the data from the other days - filming the fish is quite quickly done, but analysing the videos is an enormous amount of work for what we perceive to be little gain.

- In the sprint swimming performance analysis section, I believe it would be more statistically sound to use the average of the four escape responses, rather than the fastest one.

RESPONSE: the purpose of fast starts is to measure the maximal physiological capacity for sprint. Given variation in speed between trials even when fish are startled, it is a standard procedure to measure several fast starts and then use the maximal speed. Averaging would potentially underestimate the maximal capacity, and the variation between trails (which is likely to be due to different levels of motivation) is unrelated to our hypotheses here. We now provide extra explanation and references in the text.

- Was there a relationship between the fast start speed and the U10 speed? If there is a correlation between those two variables, you could either just use one, or create a composite variable. From the paper, it is unclear to me why both of these phenotypes were measured for “sprint speed”.

RESPONSE: the relationship between fast starts and U10 speed is shown in the Supplementary Fig. S3. There are various distinct measures of sprint speed in the literature, and to gain more generality, we included the two most common ones in our study. We explain this now in the Methods section and provide a reference.

Reviewer: 2

Comments to the Author(s)

Summary

In this manuscript, Miln et al. present an experimental study in which they aimed to identify the determinants of competitive success in zebrafish. By measuring predator escape responses and competitive foraging speeds, the authors show that competitive success (defined as being the first fish to reach a piece of food) is determined by an individual's ranking in the social hierarchy rather than their ability to swim fast. These results suggest that, in gregarious species, individuals may modulate their foraging speeds based on their social status, thus trading-off the benefits and opportunity costs of group living.

Overall, this manuscript was very well written and enjoyable to read (nice job!). I have relatively few concerns about the paper, although I do see some opportunities for improvement. I have split my feedback into major and minor comments below.

Major Comments:

- Lines 135-137: "Four escape responses were analysed for each fish and the fastest speed was used in the analysis, which we refer to as 'fast start'."

Can the authors please clarify why they elected to only use the fastest escape response for each individual, and exclude the other three measurements? It seems to me that a similar analysis could be conducted that makes use of all speed measurements for each fish (for example, by including individual as a random effect), or (less preferably) by taking the average speed across the four trials. Presumably, the escape response is not perfectly repeatable, and so there is some (possibly substantial) degree of within-individual variation. Does the analysis hold if, for example, the slowest times are used instead?

RESPONSE: please see also our response to reviewer 1 above: the purpose of fast starts is to measure the maximal physiological capacity for sprint. Given variation in speed between trials even when fish are startled, it is a standard procedure to measure several fast starts and then use the maximal speed. Averaging would potentially underestimate the maximal capacity, and the variation between trials (which is likely to be due to different levels of motivation) is unrelated to our hypotheses here. We now provide extra explanation and references in the text.

- Results

Can the authors please report all the relevant statistical information to support their

results? In my view, reporting only p-values in the absence of test statistics, sample sizes, and degrees of freedom, is insufficient.

RESPONSE: we provided extra information about permutational analyses plus references: "Permutational analyses are a randomisation procedure that use the data per se for statistical inference rather than assuming underlying distributions, which makes them more parsimonious and independent from underlying assumptions [23]. The p-value in permutational analyses is calculated as the number of randomised data sets that show the same or greater effect as the measured data set, divided by the total number of permutations; the p-value therefore denotes the probability that the observed effects could be obtained randomly. Hence, permutational analyses do not have any associated test-statistic and are often preferable to frequentist statistics [24]." We also added that the sample size was 50 fish in 10 groups of five fish each, and we now give p-values and degrees of freedom (p_{df}).

Minor Comments:

- Lines 39-40: "Here we tested the alternative hypotheses that competitive success is determined by sprint performance or by social status."

Are these two hypotheses necessarily mutually-exclusive alternatives? What if, for example, the fastest swimmers have the highest social status, and so the two are actually confounded with each other (i.e., hypothesis 3 as outlined in the introduction)?

RESPONSE: we removed "alternative" from this sentence.

- Lines 44-45: "Social hierarchies in zebrafish are established within seven days..."

Can the authors clarify what this means? Within seven days of what (presumably group formation)?

RESPONSE: we added "...within seven days of their first encounter..."

- Lines 89-90: "..., and it may alter the importance of locomotor..."

Not clear what "it" is referring to here (social relationships or locomotor behaviour). Can the authors please clarify? If "it" refers to social relationships, I recommend replacing "it" with "they".

RESPONSE: we meant to refer to social relationships, and replaces "it" with "they"

- Lines 95-96: "We therefore tested the hypotheses that i)..."

Replace the plural "hypotheses" with the singular "hypothesis".

RESPONSE: corrected as suggested

- Lines 123;124: "... and photographed to determine standard length..."

I know that this is a standard procedure, but I think it might be useful to include an example photo (or examples) in the supplementary materials to show how length was measured.

RESPONSE: We now included an example picture as Supplementary Material Fig. S1

- Line 129: "... previously[10]."

Need a space before the reference.

RESPONSE: corrected.

- Lines 159-160: "A piece of fish flake food was then placed approximately 20 cm away from the perforated container. After five seconds the contained was removed..."

Is it possible the a fish's position within the holding cylinder and thus (Euclidean) distance to the piece of food prior to the container being removed affected which fish got the food?

That is, do fish who start the trial closer to the food have a higher likelihood of getting the food (i.e., not because they are better competitors per se)? Of course, repeated sampling (10 trials per group per day) should randomize fish's starting positions, so perhaps this is not a concern.

RESPONSE: we don't think that this was a problem because of the repetition as the referee points out, and because fish did not sit still in any particular starting position.

- Lines 170-172: ""

Similar to my comment above about escape responses, I'm slightly unclear why the maximum speed was used as a proxy for realized speed here. Can the authors please clarify? It seems to me that average speed would be a much more suitable estimate of realized speed (i.e., if speed is normally distributed, the time it takes to travel between two points is much more dependent on the average speed than short-duration bursts of maximum speed that occur at the extremes). Although, perhaps given the relatively short distances being measured the average and maximum are similar to each other?

RESPONSE: We now included an additional regression analysis comparing average realised speed attained during competitive trials to maximum realised speed (now Supplementary Fig. S2) - the two are highly correlated ($p < 0.0001$) and their relationship did not differ between days ($p = 0.69$). However, we retained maximum speed as a representation of realised speed because it provides a better comparison with sprint speed (hypothesis ii), which represents maximum physiological capacity.

- Lines 207-208: "However, realised speed increased significantly ($p < 0.0001$) from day 1 to day 7."

I recommend re-structuring this sentence (and others in the results) to refocus the results on the biology and less on arbitrary conclusions about significance/non-significance. For example, what was the effect size or average increase in speed between days 1 and 7? For example: "Between days 1 and 7, realized speeds increased by an average of XX m/s (test-stat, df, $p < 0.0001$)."

RESPONSE: we changed the text as suggested.

- Lines 251-252: "It is interesting that animals rarely perform maximally under natural or naturalistic situations, which is likely to be the result of trade-offs with other fitness-related traits."

Is it possible that fish with different maximum capacities adjusted their realized speeds in different ways? From a competitive perspective, it's not essential that a fish travels at its maximum speed, but that instead that it has the fastest speed among the group of competitors. Is there any indication that, for example, those fish with fast start speeds modulate their realized speed in response to the realized speeds of others in the group? In this way, fish who *can* travel fast will appear to have reduced their realized speeds so they are only slightly faster than the next fastest fish. In contrast, fish who have slower start speeds might express relatively similar realized speeds because they need to travel near their maximum in order to compete (i.e., a conditional correlation).

RESPONSE: we explored this possibility by regressing the difference between fast-start and realised speeds against fast start speed to test whether fish with greater fast start speed show a greater reduction to realised speed. This was indeed the case, and we now included a Figure with these data as Supplementary Fig. S3 and comment in the Discussion that fish seem to conform to the group speed regardless of their physiological capacity (i.e., fast start speed).

- Lines 254-255: "Movement while competing may be interpreted within a conceptual framework that is similar to Economic Escape Theory."

Relatedly, perhaps similar theory also explains the lack of a significant correlation between maximum escape speed (in response to predator stimulus) and realized speed (in response to food). The outcomes of these two contexts/scenarios are fundamentally unbalanced (i.e., the very serious risk of death vs. the not-so-serious risk of missing out on a single foraging opportunity; "[swimming] for your life vs. [swimming] for your dinner").

RESPONSE: we now comment on the relative strength of the stimuli to escape predators vs obtaining food .

Reviewer: 3

Comments to the Author(s)

The study is an investigation of the relationship between traits related to physiological capacity as swimming performance and competitive success in zebrafish. It deals with the premise that social ranking can impact competitive success too and the study indicates that physiological performance decreases in importance as social relationships are established. While this is a relevant and interesting question to address, the study itself seems more preliminary at this stage. The conclusions drawn from the results are not satisfactory, as these are not currently sufficiently supported by the analysis and results presented. Details provided in the methods section are rather insufficient and the statistical analyses also need to be improved.

It is well known that biological factors such as the sex of individuals within groups can be important in the development of social ranking. In zebrafish too, studies have shown the role of sex in the development of dominance (e.g. Paull et al. 2010; Dahlbom et al. 2012).

This has not been addressed or discussed in the study. The results should be discussed in the context of what is known across other similar group-living species.

RESPONSE: we now clarify in the Methods that experimental fish were of mixed sex (with two or three fish of each sex per group) to provide a more natural context and avoid bias from using a single sex only, and we provide a reference commenting on the potential bias of single sex studies (Woittowich et al 2020 eLife). We now also point out that in zebrafish both males and females engage in dominance behaviour citing Paull et al. 2010.

The manuscript needs major revision before it can be recommended for acceptance. Specific comments are provided below:

Material and Methods: Line 111. How many fish were kept in each tank? How long were they kept there before starting experiments?

RESPONSE: we added that fish were kept at a density of 2-3 fish per litre for 1-2 weeks before experiments

Line 121. How do you know 3 days are enough for fish to recover after elastomer tagging? How was it ensured that elastomer tagging did not affect stress levels in individuals? Perhaps it would help to have control tests where individuals with and without tags were compared for their sprint performance or the same individuals can be tests for sprint performance before and after tagging?

RESPONSE elastomer tagging is a common procedure for small fish. We did not determine empirically whether elastomer tagging influenced swimming performance after three days, but observations of fish indicated that fish behaved normally soon after tagging (i.e. fed normally, engaged in social behaviour, and spent most of their time cruising near the water surface). We comment on this now in the methods.

Line 150. How many individuals were tested for their sprint performance? Were they all males, all females, or both?

RESPONSE: we now added that we measured swimming performance in all experimental fish (i.e. 10 groups of five fish each). Please see above regarding the sex of fish.

Line 153. As mentioned above, what was the sex of the individuals' groups? Where these mixed-sex groups?

RESPONSE: please see our response above.

Line 181. "system of? " Not clear. Please rewrite.

RESPONSE: we changed "system of [17]" to "method described in [17]"

Line 185. it is well believed that there are differences in aggressive interactions when there are same-sex or mixed-sex groups.

So it is important to address the role of the sex of individuals to understand the formation of social hierarchy.

RESPONSE: please see our response above

Statistical Analysis: Line 194. Was a random factor included in the model? As the same fish are tested across days, fish id would be used as a random factor in the model.

RESPONSE: we used "tank" as a random factor in the statistical models.

Results: Lines 204-208. It is difficult to understand the results of the study in the current version of the manuscript. Please provide sample sizes or df for your statistical analyses. Estimate values for the model should also be given.

RESPONSE: we now added more information about permutational analyses: "Permutational analyses are a randomisation procedure that use the data per se for statistical inference rather than assuming underlying distributions, which makes them more parsimonious and independent from underlying assumptions {Drummond:2012hp}. The p-value in permutational analyses is calculated as the number of randomised data sets that show the same or greater effect as the measured data set, divided by the total number of permutations; the p-value therefore denotes the probability that the observed effects could be obtained randomly. Hence, permutational analyses do not have any associated test-statistic and are often preferable to frequentist statistics {Ludbrook:1998vc}." We restated the sample size in the Statistical analysis section now (50 fish in 10 groups of five fish), and sample sizes are also given in Figure captions.

Line 210. "Realised speed and social score predicted competitive success"
Results need to be reported with the estimate, df, etc. of the model

RESPONSE: please see our response above regarding permutational analyses, and we now give df with p-values.

Line 216. What are the model R square, df, etc. for these analyses? Based on the Figure provided, the relationship is a weak positive one.

RESPONSE: regression equations and R square values are given in the Figure captions

Discussion: Line 223. This statement is not convincing, as the results presented do not necessarily prove social hierarchy as being the most important determinant of competitive success. The figures also do not show a strong enough relationship. This statement should be rewritten.

RESPONSE: We disagree: there was a significant effect of social score on competitive success, and the effect size of social score was greater than that of realised speed. We refer to effect sizes (presented in Fig. 2d) more explicitly now in the Results.

Line 228-230. Ending a sentence with “for example” sounds strange. This statement needs to be reworded.

RESPONSE: we deleted "for example"

Line 262-266. This needs to be checked, because, details of the results are not provided and the figure at least only seems to show a weakly positive relationship.

RESPONSE: We reworded this sentence and made it more conditional, clarifying that this is not a statement of results but a possible explanation.